



# Insight into PM₂.₅ Sources by Applying Positive Matrix Factorization (PMF) at an Urban and Rural Site of Beijing

**Deepchandra Srivastava[1], Jingsha Xu[1], Tuan V. Vu[1,2]**

**Di Liu[1,3], Linjie Li[3], Pingqing Fu[4], Siqi Hou[1], Zongbo Shi[1,*]**

**and Roy M. Harrison[1†*]**

**[1] School of Geography Earth and Environmental Science, University of Birmingham, Birmingham, B15 2TT United Kingdom**

**[2] Now at: School of Public Health, Imperial College London, London United Kingdom**

**[3] Institute of Atmospheric Physics, Chinese Academy of Sciences Beijing, 100029, China**

**[4] Institute of Surface-Earth System Science, Tianjin University, Tianjin 300072, China**

**† Also at: Department of Environmental Sciences / Centre of Excellence in Environmental Studies, King Abdulaziz University, PO Box 80203, Jeddah, 21589, Saudi Arabia**

**Corresponding author:** E-mail: r.m.harrison@bham.ac.uk (Roy M. Harrison) and z.shi@bham.ac.uk (Zongbo Shi)


## ABSTRACT

This study presents the source apportionment of $PM_{2.5}$ performed by PMF on data collected at an urban (Institute of Atmospheric Physics - IAP) and a rural site (Pinggu-PG) in Beijing as part of the Atmospheric Pollution and Human Health in a Chinese megacity (APHH-Beijing) field campaigns. The campaigns were carried out from 9th November to 11th December 2016 and 22nd May to 24th June 2017. The PMF included both organic and inorganic species, and a seven-factor output provided the most reasonable solution for the $PM_{2.5}$ source apportionment. These factors are interpreted to be traffic emissions, biomass burning, road dust, soil dust, coal combustion, oil combustion and secondary inorganics. Major contributors to $PM_{2.5}$ mass were secondary inorganics (22-24%), biomass burning (30-36%), and coal combustion (20-21%) sources during the winter period at both sites. Secondary inorganics (48%), road dust (20%) and coal combustion (17%) showed the highest contribution during summer at PG, while $PM_{2.5}$ particles were mainly composed of soil dust (35%) and secondary inorganics (40%) at IAP. Despite this, factors that were resolved based on metal signatures were not fully resolved and indicate a mixing of two or more sources. PMF results were also compared with sources resolved from another receptor model (i.e. CMB) and PMF performed on other measurements (i.e. online and offline aerosol mass spectrometry (AMS)) and showed a good agreement for some but not all sources. The biomass burning factor in PMF may contain aged aerosols as a good correlation was observed between biomass burning and oxygenated fractions ($r^2$=0.6-0.7) from AMS. The PMF failed to resolve some sources identified by the CMB and AMS, and appears to overestimate the dust sources. A comparison with earlier PMF source apportionment studies from the Beijing area highlights the very divergent findings from application of this method.

**Key words:** Source apportionment; $PM_{2.5}$; Beijing; PMF; CMB; online AMS; offline AMS



## 1. INTRODUCTION

Atmospheric particulate matter (PM) is composed of various chemical components and can affect air quality (and consequently human health), visibility, and ecosystems (Boucher et al., 2013; Heal et al., 2012). Through absorption and scattering of solar radiation and by affecting clouds, PM also have a major impact on the climate, and thus the hydrological cycle. PM with an aerodynamic diameter less than 2.5 μm ($PM_{2.5}$) is given special attention due to its adverse effects on human health as it can penetrate deep into human lungs when inhaled. Several recent studies have indicated that many adverse health outcomes, such as respiratory and cardiovascular morbidity and mortality, are related to long-term exposure to PM (Lu et al., 2021; Wang et al., 2016; Xing et al., 2016; Xie et al., 2019). In addition, over a million premature deaths per year are reported in China due to poor air quality (GBD MAPS Working Group, 2016). Beijing, the capital city of China, is a megacity with approximately 21 million inhabitants that are regularly exposed to severe haze events. For example, 77 pollution episodes (defined as two or more consecutive days where the average $PM_{2.5}$ concentration exceeds 75 μg m$^{-3}$) were observed between April 2013 to March 2015 (Batterman et al., 2016). $PM_{2.5}$ concentrations have reached 1,000 μg m$^{-3}$ in some heavily polluted areas of Beijing (Ji et al., 2014). In addition, a study compared the number of cases of acute cardiovascular, cerebrovascular, and respiratory diseases in the Beijing Emergency Center and haze data from Beijing Observatory between 2006 and 2013. Their results showed a rising trend, highlighting the average number of cases per day for all three diseases was higher on hazy days than on non-hazy days (Zhang et al., 2015a). Therefore, major control measures were implemented to reduce $PM_{2.5}$ pollution in Beijing (Vu et al., 2019). Recently, one-third of Chinese cities in 2020 were kept under lockdown to prevent the transmission of COVID-19 virus, which strictly curtailed personal mobility and economic activities. The lockdown led to an improvement in air quality and managed to bring down the levels of $PM_{2.5}$. Despite these improvements, $PM_{2.5}$ concentrations during the lockdown periods remained higher than the World Health Organization recommendations, suggesting much further effort is



needed (He et al., 2020; Le et al., 2020; Shi et al., 2021). A quantitative source apportionment
provides key information to support such efforts.

Receptor models are widely used for source apportionment of PM$_{2.5}$. These methods include positive
matrix factorization (PMF) (Paatero, 1997; Paatero and Tapper, 1994), principal component analysis
(PCA) (Lee et al., 2011), chemical mass balance (CMB) (Watson et al., 1990), and UNMIX (Herrera
Murillo et al., 2012). Among these methods, PMF is a widely used multivariate method that can
resolve the dominant positive factors without prior knowledge of sources. Previous PMF studies,
based on high resolution Aerosol Mass Spectrometer data, have provided valuable information on the
sources of PM in urban Beijing and its surrounding areas (Zhang et al., 2015b; Huang et al., 2010b;
Sun et al., 2010; Sun et al., 2013; Zhang et al., 2013; Zhang et al., 2014; Zhang et al., 2017; Zhang et
al., 2016; Hu et al., 2016; Qiu et al., 2019). However, the factors that influence haze formation and
related sources remain unclear due to its inherent complexity (Tie et al., 2017; Sun et al., 2014). Filter-
based PMF studies provide a valuable tool for identifying sources of airborne particles, by utilising
size-resolved chemical information (Li et al., 2019; Ma et al., 2017a; Tian et al., 2016; Yu et al.,
2013; Song et al., 2007; Song et al., 2006). These source apportionment studies have predominantly
used OC (organic carbon), EC (elemental carbon), water soluble ions and metals as the input data
matrix to explore the co-variances between species and their associated sources, but to the best of our
knowledge, the use of organic markers in PMF has not been explored extensively in Beijing. The use
of organic molecular markers in PMF has enhanced our understanding of the PM fraction as they can
be source specific (Shrivastava et al., 2007; Jaeckels et al., 2007; Zhang et al., 2009; Wang et al.,
2012; Srimuruganandam and Shiva Nagendra, 2012; Schembari et al., 2014; Laing et al., 2015;
Waked et al., 2014; Srivastava et al., 2018) and could potentially offer a clearer link between factors
and sources.





This study presents the results obtained from the PMF model applied to a filter-based dataset collected
in the Beijing metropolitan area at two sites, urban and rural. The study provides source
apportionment results from both urban and rural locations in Beijing including their temporal and
spatial variations. In addition, the study also presents a short summary of previously published filter-
based studies conducted in the Beijing metropolitan area and their major outcomes. A comparison of
the present PMF results was also made with other source apportionment approaches or applications
of PMF to other datasets, with an aim to discuss the existing PM sources in the Beijing metropolitan
area, including focusing on the strengths and weaknesses of the source apportionment approach
employed.

**2.    METHODOLOGY**
Details about the sampling site, measurements, sample collection and analytical procedures are
reported elsewhere (Shi et al., 2019; Xu et al., 2020; Wu et al., 2020), and hence only the essential
information is presented in this section.

**2.1 Sampling Site and Sample Collection**
The $PM_{2.5}$ sampling was conducted simultaneously at the urban and rural sites from 9[th] November to
12[th] December 2016 and 22[nd] May to 24[th] June 2017 as part of the Atmospheric Pollution and Human
Health in a Chinese megacity (APHH-Beijing) field campaigns (Shi et al., 2019) (Figure S1). The
urban sampling site (116.39E, 39.98N) - the Institute of Atmospheric Physics (IAP) of the Chinese
Academy of Sciences in Beijing, represents typical condition of central Beijing, there are various
roads nearby including a highway road approximately 200 m away. The rural Pinggu site (PG)
(40.17N, 117.05E) is located in Xibaidian village. This site is approximately 60 km to the north-east
of Beijing city centre and about 4 km north-west of the Pinggu town centre. The site is surrounded



by trees and farmland. In addition, residents mainly use coal and biomass for heating and cooking in
individual homes.

24-hour $PM_{2.5}$ samples were collected every day on pre-baked quartz filters (Pallflex, 8×10 inch) and
47 mm PTFE filters (flow rate of 15.0 L min$^{-1}$) using high volume (Tisch, USA, flow rate of 1.1 m$^3$
min$^{-1}$) and medium volume (Thermo Scientific Partisol 2025i) air sampler. Field blanks were also
collected during the sampling campaign at both sites. The quartz filters were then analyzed for organic
tracers, OC/EC and ion species. PTFE filters were used for the determination of $PM_{2.5}$ mass and
metals. Details on preparation and conservation of these filter samples have already been reported
elsewhere (Wu et al., 2020; Xu et al., 2020).

Real time composition of non-refractory $PM_1$ particles (NR-$PM_1$) was measured using an Aerodyne
aerosol mass spectrometer (AMS) at a time resolution of 2.5 min. The operational details on the AMS
measurements have been given elsewhere (Xu et al., 2019). In addition, the measurements of gaseous
species such as $O_3$, CO, NO, $NO_2$ and $SO_2$ were performed using gas analyzers. The meteorological
parameters including temperature (T), relative humidity (RH), wind speed (WS), and wind direction
(WD) were also measured at both sites.

**2.2     Analytical Procedure**
In all 62 and 72 chemical species were quantified in each $PM_{2.5}$ sample from IAP and PG,
respectively. This included EC/ OC, 36 organic tracers, 7 major inorganic ions ($Na^+$, $K^+$, $Ca^{2+}$, $NH_4^+$,
$Cl^-$, $NO_3^-$ and $SO_4^{2-}$) and 17 metallic elements (V, Cr, Co, Mn, Ni, Cu, Zn, As, Br, Sr, Ag, Cd, Sn,
Sb, Ba, Hg and Pb) at IAP. Similarly, the identified species at PG included EC/OC, 51 organic tracers,
7 major inorganic ions and 12 metallic elements (V, Cr, Co, Mn, Ni, Cu, Zn, As, Sr, Sb, Ba, and Pb).
EC and OC measurements were performed using a Sunset lab analyser (model RT-4) and DRI multi-


wavelength thermal-optical carbon (model 2015) analyser based on the EUSAAR2 (European
Supersites for Atmospheric Aerosol Research) transmittance protocol at both sites, IAP and PG,
respectively, following the procedure explained by Paraskevopoulou et al. (2014). Major inorganic
ions and metallic elements were analysed using an ion chromatograph (Dionex, Sunnyvale, CA,
USA) and Inductively coupled plasma-mass spectrometer (ICP-MS) at both sites, respectively. Major
crustal elements including Al, Si, Ca, Ti and Fe were determined by X-ray fluorescence spectrometer
(XRF).
Organic tracers at IAP included n-alkanes 11 $C_{24}$-$C_{34}$, 2 hopanes, 17 PAHs, 3 anhydrous sugars
(levoglucosan, mannosan, galactosan), 2 fatty acids (palmitic acid, stearic acid) and cholesterol.
These organic tracers were analysed by gas chromatography mass spectrometry (GC/MS, Agilent
7890A GC plus 5975C mass-selective detector) coupled with a DB-5MS column (30 m × 0.25 mm ×
0.25 µm) following the protocol explained in Xu et al. (2020). At PG, organic tracers were analysed
based on the method reported by Wu et al. (2020) using GC/MS (Agilent GC-6890N plus MSD-
5973N) coupled with a HP-5MS column (30 m × 0.25 mm × 0.25 µm). This included quantification
of similar species (12 n-alkanes $C_{24}$-$C_{35}$, 9 hopanes, 22 PAHs, 3 anhydrous sugars (levoglucosan,
mannosan, galactosan), 4 fatty acids (palmitic acid, stearic acid, linoleic acid, oleic acid) and
cholesterol) with few additional ones. Recoveries for the identified organic tracers ranged from 70-
100% and 80-110 %, at IAP and PG, respectively. Field blank were also analysed as part of quality
control and demonstrated very low contamination (<5%).
In addition, one or two punches of $PM_{2.5}$ filter sample were also analysed offline using AMS to
investigate the water-soluble OA (WSOA) mass spectra following the procedure explained
previously (Qiu et al., 2019).





**2.4    Positive Matrix Factorization**
Detailed information on the receptor modelling methods used within this study can be found
elsewhere (Paatero and Tapper, 1994; Hopke, 2016). Positive matrix factorization (PMF) is a
multivariate factor analysis tool and based on a weighted least square fit, where the weights are
derived from the analytical uncertainty. The best model solution was obtained by minimizing
residuals obtained between modeled and observed input species concentrations Estimation of
analytical uncertainties for the filter-based measurements was calculated using Eq. (1) (Polissar et al.,

185    1998).


$$\sigma_{ij} = \begin{cases} \frac{5}{6} LD_j & \text{if } X_{ij} < LD_j \\ \sqrt{(0.5 * LD_j)^2 + (EF_j X_{ij})^2} & \text{if } X_{ij} \geq LD_j \end{cases} \qquad \text{Eq. (1)}$$
where $LD_j$ is the detection limit for compound $j$ and $EF_j$ is the error fraction for the compound $j$. The
detection limits of all compounds used in the PMF model is given in Table S1 (SI). The U.S.
Environmental Protection Agency (US-EPA) PMF 5.0 software was used in this work to perform the
source apportionment.
***Selection of the input* data.** The selection of species used as input data for the PMF analysis is
important and can significantly influence the model results (Lim et al., 2010). The following set of
criteria were used for the selection of the input species: signal to noise ratio (S/N) (Paatero and Hopke,
2003), major PM chemical species, compounds with maximum data points above the detection limit
and those being considered as specific markers of a given source (e.g., levoglucosan, picene) (Oros
and Simoneit, 2000; Simoneit, 1999) were selected. These steps were taken to limit the input data
matrix according to the total number of samples (n=133); some species were also not included if they
belonged to a single source and correlated with another marker of this source. A total of 31 species
were used in the model (PM$_{2.5}$, OC, EC, K$^+$, Na$^+$, Ca$^{2+}$, NH$_4^+$, NO$_3^-$, SO$_4^{2-}$, Cl$^-$, Ti, V, Mn, Ni, Zn, Pb,




Cu, Fe, Al, C26, C29, C31, 17α(H)-22,29,30-trisnorhopane,17β(H),21α(H)-30-norhopane,chrysene,
benzo(b)fluoranthene, benzo(a)pyrene, picene, corene, levoglucosan, and stearic acid). The
concentration of $PM_{2.5}$ was included as a total variable in the model (with large uncertainties) to
directly determine the source contributions to the daily mass concentrations.
*Selection of the final* **solution.** As normally recommended, a detailed evaluation of factor profiles,
temporal trends, fractional contribution of major species to each factor and correlations with external
tracers, were investigated carefully to select the appropriate number of factors.
A few constraints were also applied to the base run to obtain clearer chemical source profiles in the
final solution. The general framework for applying constraints to PMF solutions has already been
discussed elsewhere (Amato et al., 2009; Amato and Hopke, 2012). The changes in the $Q$ values were
considered here as a diagnostic parameter to provide insight into the rotation of factors. All model
runs were carefully monitored by examining the $Q$ values obtained in the robust mode. To limit
change in the $Q$-value, only "soft pulling" constraints were applied. The change in the $Q$-robust was
< 1%, which is acceptable as per PMF guidelines (< 5%) (Norris et al., 2014). Finally, three criteria
were used to select the optimal solution, including correlation coefficient ($r$) between the measured
and modelled species, bootstrap and t-test (two-tailed paired t-test) performed on the base and
constraint runs, as explained previously (Srivastava et al., 2018).
**2.5      Back trajectories and Geographical Origins**
The geographical origin of selected identified sources and pollutants was investigated using
concentration-weighted trajectory (CWT), non-parametric wind regression (NWR), and cluster
analysis methods. NWR combines ambient concentrations with co-located measurements of wind
direction and speed and highlights wind sectors that are associated with high measured concentrations
(Henry et al., 2009). The general principle is to smooth the data over a fine grid, so that a weighted



concentration could be estimated by any wind direction (ɸ)/wind speed (v) couple, where the
weighing coefficients are determined through Gaussian-like functions. CWT and cluster analysis
assess the potential transport of pollution over large geographical scale (Polissar et al., 2001). These
approaches combine atmospheric concentrations measured at the receptor site with back trajectories
and residence time information and help to geographically evaluate air parcels responsible for high
concentrations. For this purpose, hourly 24-h back trajectories arriving at 200 m above sea level were
calculated from the PC-based version of HYSPLIT v4.1 (Stein et al., 2015; Draxler, 1999). NWR,
CWT calculations and cluster analysis, were performed using the ZeFir Igor package (Petit et al.,

237    2017).


### 239 2.6 Other Receptor Modelling Approaches

Sources were also resolved at both sites separately using another receptor model known as chemical
mass balance (CMB) as well as PMF performed on high resolution AMS data collected at IAP. Details
on sources resolved using these approaches are reported elsewhere (Wu et al., 2020; Xu et al., 2020;
Sun et al., 2020).

Briefly, CMB is based on a linear least squares approach and accounts for uncertainties in both, source
profiles and ambient measurements to apportion the sources of OC. The US EPA CMB8.2 software
was used for this purpose at both sites. The source profiles applied in the model were taken from local
studies to better represent the sources, including profiles of straw burning (Zhang et al., 2007), wood
burning (Wang et al. 2009), gasoline and diesel vehicles (Cai et al. 2017), industrial and residential
coal combustion (Zhang et al., 2008), and cooking (Zhao et al., 2015). Only the source profile for
vegetative detritus (Rogge et al. 1993; Wang et al., 2009) was not available from local studies. The
selected fitting species included EC, anhydrous sugar (levoglucosan), fatty acids, PAHs, hopanes and
alkanes



## 3.    RESULTS AND DISCUSSION

### 3.1    Overview on PM Sources in Beijing based on the Current Source Apportionment Study

A seven-factor output provided the most reasonable solution for the PM$_{2.5}$ source apportionment performed on the combined dataset from IAP and PG (Figures 1 and 2).

Based on the factor profiles, we identified traffic emissions, biomass burning, road dust, soil dust, coal combustion, oil combustion and secondary inorganics. For the same dataset, solutions with six sources were less explanatory and some factors were mixed. Conversely, an increase in the number of factors led to the split of meaningful factor profiles. In the final solution, the comparison of the reconstructed PM$_{2.5}$ contributions from all sources with measured PM$_{2.5}$ concentrations for different seasons at both sites showed good mass closure ($r^2 = 0.61$-$0.91$, slope = 0.99-1.12, p < 0.05, ODR (orthogonal distance regression)). A low $r^2$ (0.61) value was observed for the summer period at IAP (Figure S2). This may be due to the inability of PMF to model low concentrations observed for sources such as biomass burning and coal combustion during the summer. In addition, most of the species showed good agreement with measured concentrations (Table S2). Bootstrapping on the final solution showed stable results with more than 95 out of 100 bootstrap mapped factors (Table S3). Finally, no significant difference (p>0.05) was observed in the factor chemical profiles between the base and the constrained runs (Table S4).

Overall, secondary inorganics, biomass burning, and coal combustion sources were the main contributors to the total PM$_{2.5}$ mass during winter (Figure 3). These sources accounted for 22%, 36%, 20%, and 24%, 30%, 21% of PM$_{2.5}$ mass at IAP and PG, respectively. Secondary inorganics, road dust and coal combustion showed the highest contribution during summer at PG, while PM$_{2.5}$ particles were mainly composed of soil dust and secondary inorganics at IAP. Identified aerosol sources, factor profiles and temporal evolutions are discussed below. Note, PMF was carried out on the combined datasets and thus only provides a single set of factor profiles for both sites. Similar to previous studies

(Li et al., 2019; Ma et al., 2017a; Tian et al., 2016; Yu et al., 2013; Liu et al., 2019; Zhang et al., 2013), neither secondary organic aerosol nor cooking emissions were identified, and given the good mass closure must be present within other source categories.

***Coal combustion***. Coal combustion was identified based on it accounting for a high proportion of PAHs (27-78%), especially picene (78%) as a specific marker of coal combustion (Oros and Simoneit, 2000), together with significant amount of OC (45%) and EC (29%) (Figure 1). This factor also made a substantial contribution to n-alkanes (28-58%), stearic acid (64%) and hopanes (53-56%), as these compounds are also abundant in coal smoke (Bi et al., 2008; Zhang et al., 2008; Oros and Simoneit, 2000; Guo et al., 2015).

The coal combustion factor accounted for 20% of the PM mass (16.0 μg $m^{-3}$) at the urban site IAP during winter and followed typical seasonal variations. However, the contributions of this source to $PM_{2.5}$ mass were broadly similar (21% vs 17%, Figure 3) at PG during both seasons, while the average concentrations were higher in winter than summer (19.4 μg $m^{-3}$ > 4.6 μg $m^{-3}$). Due to a lack of infrastructure at the rural site PG, the residents still use coal for cooking and heating purposes at the time of sampling (Shi et al., 2019). There is a reduction in coal usage for heating due to elevated temperatures in the summertime, leading to low levels of this factor. But the similar contribution at the rural site could be linked to consistent cooking activities throughout the year (Figure 2) (Shi et al., 2019; Tao et al., 2018). These results were in good agreement with previous observations reported at the same urban site (18%) (Ma et al., 2017a; Tian et al., 2016). In addition, similar contributions were also observed at other urban locations around Beijing (Wang et al., 2008; Liu et al., 2019).

This factor also included significant contributions from levoglucosan (60%). Levoglucosan, a major pyrolysis product of cellulose, and has been proposed as molecular marker of biomass burning aerosols (Simoneit, 1999). A study conducted in China suggested that residential coal combustion can also contribute significantly to levoglucosan emissions, based on both source testing and ambient





measurements (Yan et al., 2018). Therefore, it is expected that the contribution of levoglucosan is
probably linked to residential coal use for cooking in this case. It is also possible that the high
contribution of levoglucosan could also be linked to model bias as the PMF model only provides an
average factor profiles for both sites irrespective of their nature (rural vs. urban) and different
sampling periods (summer vs. winter).

This was further supported as NWR and CWT analysis showed similar results, mostly dominated by
a northerly flow during the winter period at both sites. High concentrations of this source and
levoglucosan were observed at low wind speeds (Figure S3), indicating the significant role of local
activities. Higher levels were observed at the rural site PG (19.4 µg m$^{-3}$ vs 16.0 µg m$^{-3}$ at the urban
site). However, a south-westerly flow was dominant during summer and could be related to transport
of air masses from the Hebei province where a large number of industries operate.

***Oil combustion.*** The oil combustion factor profile included high contributions to V (79%) and Ni
(70%) (see Figure 1). V and Ni are widely used markers for oil combustion in residential, commercial
and industrial applications (Viana et al., 2008; Mazzei et al., 2008; Pant et al., 2015; Huang et al.,
2021). The V/Ni ratio obtained in this study was 0.9, close to the previously obtained ratio for residual
oil used in power plants (Swietlicki and Krejci, 1996). Results suggest this source might be attributed
to residual oil combustion linked to industrial activities as a large number of highly polluting
industries are still located in the Beijing neighbourhood (Li et al., 2019). CWT and NWR analysis
suggested the influence of regional transport at both sites, highlighting the dominance of south
westerly and south easterly flows during the winter and summer at both sites (Figure S4).

The source did not show any seasonal pattern (Figure 2), and accounted for 2% (1.4 µg m$^{-3}$) and 6%
(1.6 µg m$^{-3}$) at IAP, and 8% (7.1 µg m$^{-3}$) and 9% (2.1 µg m$^{-3}$) at PG of the PM$_{2.5}$ mass during winter
and summer, respectively (Figure 3). The contribution of this source to the PM mass was within the





similar range to the previous study conducted at the same urban site (contribution 4.7%) (Li et al.,
2019), which also found a high proportion of V attributed to the identified source.

***Biomass burning.*** The biomass burning factor was characterized by high contributions to Cl⁻ (74%),
$K^+$ (27%) and levoglucosan (25%) (Figure 1). This factor also made significant contributions to PAHs
(Chry (66%), B[b]F (66%), Cor (68%)) and followed a clear seasonal variation with a higher
contribution in winter (Figure 2). It accounted for 36 % (29.0 µg m⁻³) and 30% (27.3 µg m⁻³) of the
PM$_{2.5}$ mass during the wintertime at IAP and PG, respectively (Figure 3), while the contribution
during the summertime was extremely low. This was expected due to elevated temperature during
the summer period and reduction in biomass burning activities. In addition, $NO_3^-$(24%), and $NH_4^+$
(24%) also contributed significantly to the biomass burning factor. Biomass burning is an important
natural source of $NH_3$ (Zhou et al., 2020) which rapidly reacts with $HNO_3$ to form $NH_4NO_3$ aerosols.
The presence of $NH_4NO_3$ aerosols in biomass burning plumes, has also been reported previously
(Paulot et al., 2017; Zhao et al., 2020).

It was unexpected to observe a low contribution of levoglucosan, a known biomass burning marker,
to this factor. However, model bias and the contribution of other relevant sources to levoglucosan
could cause such observations, as discussed above.$K^+$ is also produced from the combustion of wood
lignin and has been used extensively as an inorganic tracer to apportion biomass burning contributions
to ambient aerosol (Zhang et al., 2010; Lee et al., 2008b). However, the contribution of $K^+$ to this
factor was not relatively low, possibly because $K^+$ has also other sources, such as soil dust (Duvall et
al., 2008). Cl⁻ can be emitted from both coal combustion and biomass burning, especially during the
cold period in Beijing (Sun et al., 2006). It is also important to note that high Cl⁻ levels observed in
this factor could be associated with coal combustion as Cl⁻ has been used to represent coal combustion
activities in China (Wang et al., 2008). If we consider this, high Cl⁻ levels related to the coal
combustion factor should have also shown a significant contribution to PM mass during the



summertime at the rural site (PG) as residents near the rural site mostly use coal and biomass for
cooking activities as discussed above, but they do not. Results suggest this factor can be attributed
mainly to biomass burning aerosols in the Beijing metropolitan area, but some influence of coal
combustion signals cannot be ignored. Back trajectory analysis also confirmed the local origin of this
source during the wintertime at both sites (Figure S5).

The source contribution reported in the present study was higher than that found in earlier studies in
Beijing (11-20%) (Li et al., 2019; Ma et al., 2017a; Tian et al., 2016; Yu et al., 2013; Song et al.,
2007; Song et al., 2006; Liu et al., 2019), suggesting some inclusion of coal burning. As both these
sources follow a similar typical seasonal variation, i.e., high concentration during the cold period, it
makes their separation difficult due to correlation.
***Secondary inorganics***.  Secondary inorganics were typically characterized by high contributions to
$NO_3^-$, $SO_4^{2-}$ and $NH_4^+$ (55%, 56% and 56% of the total species mass, respectively) (Figure 1). This
factor showed a temporal variation, with remarkably high concentrations observed during the period
of high relative humidity (RH) and low ozone concentration in the winter (Figure S6). The
heterogeneous reactions on pre-existing particles in the polluted environment under high RH and low
ozone conditions have been shown to play a key role in the formation of secondary aerosols compared
to gas-phase photochemical processes (Sun et al., 2013; Niu et al., 2016; Ma et al., 2017b). Therefore,
aqueous phase processes may be the major formation pathway for secondary inorganic aerosols in
Beijing during the study period. Additionally, the factor showed a similar contribution (22-24%) to
PM mass in winter at both sites. This value is lower than the value reported at the other urban location
(44%) in Beijing as a part of same campaign (Liu et al., 2019), although it should be noted that the
sampling site and dates differed.





The highest contribution to the PM mass was observed during the summertime, with an average
concentration of 11.1 µg m$^{-3}$ (40%) and 13.2 µg m$^{-3}$ (48%) at IAP and PG, respectively (Figure 3).
***Traffic emissions.***  The traffic emissions factor showed relatively high contributions to metallic
elements, such as Zn (47%), Pb (57%), Mn (27%) and Fe (22%) (Figure 1). Zn is a major additive to
lubricant oil. Zn and Fe can also originate from tyre abrasion, brake linings, lubricants and corrosion
of vehicular parts and tailpipe emission (Pant and Harrison, 2012; Pant and Harrison, 2013;
Grigoratos and Martini, 2015; Piscitello et al., 2021). As the use of Pb additives in gasoline has been
banned since 1997 in Beijing, the observed Pb emissions may be associated with wear (tyre/brake)
rather than fuel combustion (Smichowski et al., 2007). These results suggest the contribution of both
exhaust and non-exhaust traffic emissions to this factor. Further insight on the type of non-exhaust
emissions is hard to predict as these metal concentrations vary according to several parameters, such
as traffic volume and patterns, vehicle fleet characteristics and the climate and geology of the region
(Duong and Lee, 2011).
Traffic sources accounted for 9% and 6% of PM$_{2.5}$ mass during the winter and summertime at IAP
(Figure 3), corresponding to an average concentration of 7.4 µg m$^{-3}$ and 1.8 µg m$^{-3}$, respectively. In
addition, a low contribution (4%) was observed at PG during both seasons as PG experiences a low
traffic volume. The contribution of the traffic source to the PM$_{2.5}$ mass was found to be low compared
to other studies conducted in the Beijing area; (14-20%) (Li et al., 2019; Tian et al., 2016; Yu et al.,
2013; Liu et al., 2019), with the exception of a study by Ma et al. (2017a) where a similar contribution
was reported. The observed low contribution was further supported as a recent study also confirms
that road traffic remains a dominant source of NO$_x$ and primary coarse PM, however, it only accounts
a relatively small fraction of PM$_{2.5}$ mass at urban locations in Beijing (Harrison et al., 2021). It should
be noted that nitrate that can be formed from NO$_x$ emitted from road traffic is not included in this
factor. Despite the low factor contribution, the resolved chemical profile of this source was consistent
with previously identified profiles linked to road traffic emissions in the Beijing area (Ma et al.,





2017a; Yu et al., 2013). We noticed that OC/EC contribution in this factor is relatively low, while it
may be higher in traffic emissions. However, given the modern gasoline fleet in Beijing (Jing et al.,
2016), it is not unexpected to observe low OC and EC contribution. We do expect higher OC and EC
contribution from diesel vehicles. In addition, there was no obvious seasonal variation as expected,
though slightly higher concentrations were observed in the cold period, probably due to the typical
atmospheric dynamics, and consequent poorer dispersion at this time of year.
Metallic elements such as Mn, Fe and Zn were also used previously to indicate industrial activities
(Li et al., 2017; Yu et al., 2013). Back trajectory analysis reveals the influence of local emissions with
a slight regional transport during the winter period at both sites, dominated by north easterly flow
(Figure S7). Therefore, there is a possibility that these elements could also come from the Hebei
province where a large number of smelter industries are located. North-easterly and south-easterly
flows were dominant during the summer period at IAP and PG with possible regional influence. These
observations suggest that indeed this factor contains traffic aerosols, though a significant influence
of industrial emissions cannot be ruled out.
***Road dust***.  This factor makes a major contribution to crustal species, such as $Na^+$, Al and Fe (60%,
48% and 34% of species in this factor respectively) suggesting this factor may represent the
characteristics of a dust related source as reported previously (Kim and Koh, 2020). In addition, the
given factor also included significant contributions to Mn, Pb and Zn (26%, 23% and 20% of species
in this factor respectively), which are associated with brake and tyre wear as mentioned above (Pant
and Harrison, 2012; Pant and Harrison, 2013; Grigoratos and Martini, 2015; Piscitello et al., 2021).
High concentrations of Zn and Pb have also been reported for particles emitted from asphalt pavement
(Canepari et al., 2008; Sörme et al., 2001). In addition, the ratio of Fe/Al observed in the factor
chemical profile was 1.26, much higher than the value observed in the earth's crust (0.6), suggesting
an anthropogenic origin of some Fe (Sun et al., 2005). This is likely as processes associated with
vehicles, such as tyre/brake wear and road abrasion, can contaminate soil with metals, as the urban

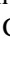

sampling site is located close to roads suggesting the resolved factor is likely linked to road dust
emissions. These metals (Fe and Al) can also have industrial sources as already reported in the Beijing
area (Wang et al., 2008; Tian et al., 2016; Yu et al., 2013; Li et al 2019). The Beijing-Tianjin-Hebei
region is the largest urbanised megalopolis region in northern China, and home to many iron and steel
making industries. Fe is characteristic components of iron and steel industry emissions (Li et al.,
2019) while Al may also come from metal processing (Yu et al., 2013). However, disentangling the
influence of industrial emissions would require further investigation.
This source also made significant contributions to OC, EC and $SO_4^{2-}$ (11-19%) (Figure 1) and was
consistent with the road dust source profiles observed previously in the Beijing area (Song et al.,
2006; Song et al., 2007; Tian et al., 2016; Yu et al., 2013). This factor accounted for 20% of the $PM_{2.5}$
mass during the summertime (5.5 µg m$^{-3}$) with exceptionally low contribution (3%) during the cold
period at PG (Figure 3). However, the factor contribution at IAP was similar during both seasons. In
addition, the contribution to PM mass at IAP in this study was similar to that reported by Tian et al.
(2016) and the studied urban site in both cases was the same. Crustal dust mass was also estimated
based on the concentrations of Al, Si, Fe, Ca, and Ti using the equation below (Chan et al., 1997).

454         $$\text{Crustal dust} = 1.16(1.9\text{Al} + 2.15\text{Si} + 1.41\text{Ca} + 1.67\text{Ti} + 2.09\text{Fe})$$

Good correlation was observed between the estimated crustal dust and this factor during both seasons
at PG (rural, winter: $r^2 = 0.78$, m (slope) =0.9; summer: $r^2 = 0.94$, m=0.5) and IAP (urban, winter: $r^2$
$= 0.51$, m=1.3; summer: $r^2 = 0.68$, m=1.2), highlighting that this may also contain a significant
fraction of crustal dust (Figure S8). This suggests the identified factor is not resolved cleanly and
contains a mixed characteristic of road dust and crustal dust.
***Soil dust.*** This factor mainly represents wind-blown soils and was typically characterized by a high
contribution to crustal elements, such as Ti (63%), $Ca^{2+}$ (41%), Fe (27%) and Al (17%) (Figure 1).





In addition, the contributions to Mn and Zn in the factor profile (Mn=24%, Zn=15%) suggest that the
given source also included resuspended road dust but probably to a lesser extent. No clear seasonal
variation was observed. This source also showed a significant contribution to n-alkanes (e.g., C29,
C31), derived from epicuticular waxes of higher plant biomass (Kolattukudy, 1976; Eglinton et al.,
1962), with the highest contribution (37%) to C31. This suggests the presence of plant derived organic
matter in the soil dust, which is also consistent with a high contribution to OC (15%).

The factor showed a high contribution (35%, 9.8 µg m$^{-3}$) to PM$_{2.5}$ mass during the summertime at
IAP, while the contribution during other seasons at both sites was less than 10% (Figure 3). The factor
profile resolved here was similar to the profile reported by Ma et al. (2017a) for soil dust, but their
soil dust factor only showed a 10 % contribution to PM$_{2.5}$ mass. In addition, other previous studies
(Yu et al., 2013; Zhang et al., 2013) also reported significant contribution of soil dust to PM$_{2.5}$ mass,
suggesting that soil dust is an important contributor to PM$_{2.5}$ mass in the Beijing area. It is also
expected as Beijing is in a semi-arid region and there are sparsely vegetated surfaces both within and
outside the city. This factor also showed good agreement with the crustal fraction estimated from the
element masses only during winter at PG ($r^2$ =0.51) and summer at IAP ($r^2$ =0.58). This again
highlights the probable mixing of this source with other factors, or mis-attribution. Back trajectory
analysis also indicates the influence of regional transport during the summer period at IAP, dominated
by south easterly-westerly flow (Figure S9) due to high windspeeds (3.6 m s$^{-1}$). Therefore, there is a
possibility that the high contribution is linked to long-range transport in advected air masses. A recent
study (Gu et al., 2020) conducted in Beijing showed the high concentrations of more oxidised aerosols
during summer due to enhanced photochemical processes; however, such type of source was not
resolved due to a lack of filter based markers. This suggests the given source may contain some
influence from an unidentified/unresolved SOC fraction. Although the most plausible attribution
appears to be to soil dust, it is not fully resolved from other sources.



The use of Si in PMF could provide a better understanding on these dust related sources. However, it
is not used in the present PMF input due to high number of missing data points. The sensitivity of
PMF results to the use of Si has also been investigated by adding Si to the input matrix and providing
high uncertainty to the missing data. No change was observed in the factor profile and temporal
variation of the resolved factors compared to the present one. In addition, we also noticed a good
correlation between Si and Al, where Al has been used in PMF (Figure S10). Several PMF runs were
also made with inorganic data only, however the resolved factors were either mixed or hard to
identify. In addition, attempts to improve the PMF results by varying the input species and by
analysing data for the IAP and PG sites separately did not offer any advantage.
**3.2    Comparison of Filter-Based PMF Results with other Receptor Modelling Approaches**
**on the same Dataset**
The source apportionment results from PMF were compared with those from CMB on the same filter-
based composition data and PMF performed on other measurements (i.e. online AMS ($PM_1$), offline
AMS ($PM_{2.5}$)) to get a deeper insight into the identified PMF factors and their origins (Figs. 4, 5, 6,
7). The CMB method resulted in the estimation of eight OC sources (i.e., vegetative detritus,
residential coal combustion (CC), industrial CC, cooking, diesel vehicles, gasoline vehicles, biomass
burning, other OC), including one secondary factor (Other OC) at both sites (IAP and PG) The online
AMS datasets allowed the identification of 6 OA (MOOOA (more oxidised oxygenated OA),
LOOOA (low more oxidised oxygenated OA), OPOA (oxidised primary OA), BBOA (biomass
burning OA), COA (cooking OA), CCOA (coal combustion OA) factors during winter at IAP, while
analyses on the offline AMS measurements resolved 4 OA (OOA, BBOA, COA, CCOA) factors.
For these analyses, OC concentrations related to the online/offline AMS OA factors were further
calculated by applying OC-to-OA conversion factors specific to each source, i.e., 1.35 for coal
combustion organic carbon (Sun et al., 2016), 1.38 for cooking organic carbon, 1.58 for biomass





burning organic carbon (Xu et al., 2019), and 1.78 for the oxygenated fraction (Huang et al., 2010a)
and used to evaluate the OC concentrations of relevant OA factors.

Only OC equivalent concentrations were used to perform comparison for all approaches. OC mass
closure was also verified at IAP during the wintertime by investigating the relation between: OC
modelled by online AMS PMF vs filter based PMF ($r^2$=0.7, slope=1.17), OC measured vs OC
modelled by filter based PMF ($r^2$=0.7, slope=1.07), OC measured vs OC modelled by online AMS
PMF ($r^2$=0.9, slope=0.92), OC modelled by offline AMS PMF vs OC model by filter based PMF
($r^2$=0.6, slope=0.75), OC measured vs OC modelled by offline AMS PMF ($r^2$=0.9, slope=1.41), and
OC measured vs WSOA (offline AMS) ($r^2$=0.9, slope=0.85) (Figure S11). The comparison of OC
modelled by PMF and CMB was also investigated at IAP ($r^2$=0.8, slope=1.05) and PG ($r^2$=0.6,
slope=1.78) (Figure S12). All source apportionment approaches showed a fairly good agreement in
reconstructing the total OC mass, justifying their direct comparison. In addition, it should be noted
that the difference in the sampling size cut-off between online AMS (NR-PM$_1$) and filter
measurements (PM$_{2.5}$) may contributes to the differences observed in the source apportionment
results.   Therefore, we also compared the relation between NR-PM measured vs PM measured
($r^2$=0.96, slope=0.92), and NR-PM measured vs PM modelled by filter based PMF ($r^2$=0.9,
slope=1.29) (Figure S13).  The agreements observed suggests that the most of the PM$_{2.5}$ mass was
accounted for by the PM$_1$ fraction, indicating that the difference in the size-cut off is relatively small.

**(a)       With CMB results at IAP**
Resolved CMB and PMF factors were compared including data from both seasons at IAP and PG
(Figure 4). A good correlation ($r^2$=0.6, n=68, p<0.05) was observed between biomass burning factors,
suggesting that this source was well resolved using both approaches (Figure 4). However, a slightly
higher concentration was reported by the CMB model (2.0 and 1.6 μg m$^{-3}$ by CMB and PMF
respectively). Individual coal combustion factors (industrial/residential) did not shown any





significant correlation ($r^2<0.2$) with the coal combustion factor identified using PMF, although the
total coal combustion fraction from CMB, the sum of industrial and residential fractions, did show
an improved correlation ($r^2=0.4$). Some improvement on the correlation was seen if two outlier
datapoints were removed (see Figure 4). A likely reason is that PMF did not resolve coal combustion
and biomass burning factors well as both factors presented a strong seasonal pattern with high
concentrations during the winter. Another possibility is the difficulty in resolving primary and
secondary fractions due to a lack of secondary organic markers used in the study. This was further
supported by the fact noted above that the PMF biomass burning factor also contained some signal
from coal combustion activities. The sum of coal combustion and biomass burning factors from both
approaches showed a good correlation ($r^2=0.7$, n=68, $p<0.05$), suggesting a common emission pattern
(e.g., high in winter and low in summer), making it challenging to resolve them. Factors linked to
vehicle emissions did not show any correlation. A weak correlation ($r^2=0.3$, n=68, $p<0.05$) was
observed between Other OC from CMB, a proxy for the secondary organic fraction and the PMF
secondary inorganics factor. In addition, other OC also weakly correlated with soil dust ($r^2=0.22$,
n=34, $p<0.05$) in summer, suggesting the mixing of unresolved secondary fraction with soil dust
profile and supports the hypothesis discussed above. It should be noted that other OC could also
contain unresolved primary fractions as PMF results indicated substantial influence of industrial
emissions and dust related sources. However, the source profiles related to industrial emissions and
dust  were not accounted for in the CMB model (Xu et al., 2020).

**(b)      With CMB results at PG**
The comparison was also made using data from both seasons at PG (Figure 5). Biomass burning
aerosols showed a good correlation for both approaches ($r^2=0.7$, n=20, $p<0.05$) but a substantially
higher concentration was estimated by the CMB model (5.1 µg m$^{-3}$ and 2.0 µg m$^{-3}$ by CMB and PMF
respectively). A significant correlation was also seen between traffic related factors from CMB and





PMF (gasoline-CMB vs traffic ($r^2$=0.6, n=20, p<0.05), diesel-CMB vs traffic ($r^2$=0.6, n=20, p<0.05)),
indicating that traffic sources resolved using PMF at the PG site may have included signals from both
diesel and gasoline vehicles; however it was not conclusive at the IAP site, as discussed above. This
suggests the traffic source resolved using PMF may contain particles linked to traffic emissions, but
the influence of other sources is prominent at IAP and resulted in poor correlation. In addition, for
traffic related factors from CMB, both showed a higher concentration (gasoline-CMB=0.8 µg m$^{-3}$,
diesel-CMB=4.5 µg m$^{-3}$, traffic-PMF=0.2 µg m$^{-3}$). As with IAP, no significant correlation was
observed between coal combustion factors from both approaches. The sum of coal combustion and
biomass burning factors from both approaches also did not present a good correlation ($r^2$=0.3, n=20,
p<0.05). This highlights the limitation of these methodologies to apportion sources when extreme
meteorological conditions may lead to high internal mixing of sources. Unfavourable dispersion
conditions have been previously observed in the Beijing region during severe haze events in winter
(Wang et al., 2014). A high correlation was observed between Other OC (CMB) and secondary
inorganics (PMF) ($r^2$=0.7, n=20, p<0.05). In addition, Other OC also showed a very high correlation
with the biomass burning factor resolved from PMF ($r^2$=0.9, n=20, p<0.05). This suggests that the
biomass burning factor in PMF may contain a substantial amount of aged aerosols since carbon
emitted during biomass burning is in some cases oxygenated and water soluble (Lee et al., 2008a)
and is subject to rapid oxidation in the atmosphere.

**(c)      With online AMS PMF factors at IAP (winter)**
BBOC (biomass burning OC) from PMF-AMS analysis agreed well with that from PMF ($r^2$=0.7,
n=27, p<0.05; 4.0 µg m$^{-3}$ and 3.1 µg m$^{-3}$ by online AMS and PMF, respectively) (Figure 6). Coal
combustion related factors showed a modest correlation (CCOC (coal combustion OC) vs coal
combustion-PMF, $r^2$=0.4, n=27, p<0.05) but the mass concentration of the coal combustion source
by PMF (11.3 µg m$^{-3}$) is significantly higher than by PMF-AMS (CCOC=4.7 µg m$^{-3}$). This may
partly be due to the different size cut offs used by these measurements (PM$_1$ for AMS vs PM$_{2.5}$). In



addition, significant improvement on the correlation was seen if two outlying points were removed
($r^2$=0.8, see Figure 6). Oxygenated fractions from AMS, MOOOC (more oxidised oxygenated OC)
and LOOOC (low oxidised oxygenated OC) also exhibited a good correlation with secondary
inorganics (LOOOC vs secondary inorganics ($r^2$=0.6, n=27, p<0.05, LOOOC=2.9 µg m$^{-3}$, secondary
inorganics=1.6 µg m$^{-3}$), MOOOC vs secondary inorganics ($r^2$=0.7, n=27, p<0.05, MOOOC=4.4 µg
m$^{-3}$)). This was also confirmed by LOOOC and MOOOC showing a good correlation with $NO_3^-$ and
$SO_4^{2-}$ previously (Cao et al., 2017). This suggests the oxygenated fractions from AMS and secondary
inorganics are subject to similar controls in the atmosphere. In addition, both oxygenated fractions
were also found to be correlated with biomass burning aerosols (LOOOC vs biomass burning-PMF
($r^2$=0.7, n=27, p<0.05), MOOOC vs biomass burning-PMF ($r^2$=0.6, n=27, p<0.05)). This further
highlights a potentially important role of biomass burning activity in SOA formation at IAP. A good
correlation was also observed between OPOC (oxidised primary OC) and secondary inorganics and
biomass burning ($r^2$=0.7, n=27, p<0.05).

**(d)     Offline AMS PMF factors at IAP (winter)**
BBOC from PMF-offline AMS analysis showed a good correlation with that from PMF ($r^2$=0.6, n=32,
p<0.05) (Figure 7) but the mass concentration of BBOC (4.6 µg m$^{-3}$) is higher than biomass burning
(3.1 µg m$^{-3}$) from PMF. This was also noticed above while comparing with BBOC resolved using
online AMS PMF, suggesting a potential uncertainty in estimating the source contribution from
biomass burning. The uncertainty in filter-based PMF analysis could be related to model error. This
was further supported as biomass burning factor also made significant contributions to $Ca^{2+}$ (15%),
Ni (30%), Cu (50%), and Al (35%), and these species are not necessarily from biomass burning
emissions but they were not resolved by PMF. In addition, the uncertainties linked to PMF-AMS
analysis could also contribute. A high correlation was noticed for secondary factors resolved using
both approaches (OOC (oxygenated OC) vs Secondary inorganics, $r^2$=0.8, n=32, p<0.05). OOC also
showed a good correlation with the biomass burning factor (OOC vs biomass burning-PMF, $r^2$=0.7,





n=32, p<0.05). This supports the hypothesis discussed previously on the origin of oxygenated
fractions.
Overall, the comparison of filter based PMF results was in broad agreement with other receptor
modelling approaches applied on the same dataset. However, large discrepancies were also observed
for some factors / sources. Common sources such as biomass burning and coal combustion were well
resolved using all approaches with some exceptions observed when using filter based PMF approach.
This could be linked to internal mixing of sources when the influence of climate and local
meteorology on both sources is predominant and making it challenging to resolve using PMF. A good
agreement was also observed between secondary inorganic aerosols and secondary fractions resolved
using other approaches. However, sources identified based on metal signatures using PMF indicated
some mixing or mis-attribution. For example, the influence of unresolved SOC on the soil dust profile
was observed during summer.
### 633    3.3.      Comparison with Previous PMF Source Apportionment Results in Beijing

In this section an attempt has been made to understand the PM sources identified in the Beijing
metropolitan area by previous studies. The goal was here to assess the previous PMF source
apportionment results and report any discrepancies noticed in the resolved sources using PMF. This
may provide useful insight on sources resolved in the present study and also in exploring the issues
associated with filter based PMF modelling in the Beijing metropolitan area. Details of the studies
conducted to evaluate PM sources using a PMF model applied to inorganic and organic markers in
the Beijing metropolitan area are presented in Table 1 and the major outcomes are discussed hereafter.
Overall, these previous PMF studies provide insights on PM sources in the Beijing metropolitan area
(Li et al., 2019; Ma et al., 2017a; Tian et al., 2016; Yu et al., 2013; Song et al., 2007; Song et al.,
2006; Liu et al., 2019; Wang et al., 2008; Zhang et al., 2013). The major identified sources are dust,





traffic emissions, coal combustion, industrial activity, secondary inorganic aerosols and biomass
burning. Although there is a general issue of their inability to identify sources such as secondary
organic aerosol and cooking emissions, similar to the present study, due to the lack of organic markers
used in the PMF model to apportion these sources. However, beyond this, their PMF outcomes were
not consistent. Large discrepancies between the sources were seen (Table 1) based on the sources
identified as well as their contribution to PM mass concentrations. Several factors could cause these
differences such as the chemical species used as input in the PMF model, the period of the study,
identification of sources based on chemical signatures and changes to the sources with time.
Input species considered within the previous studies were combinations of water-soluble ions,
metallic elements, OC and EC. Similar input species were used in all of these studies, with the
exception of the studies by Yu et al. (2013) and Li et al. (2019) who used only metallic elements for
the source apportionment. As shown in this study, including organic markers may help to resolve
some of the primary sources.
Another important parameter, the chemical species used for identifying sources were not always
consistent. For example, coal combustion was resolved based on high contribution of OC, EC, and
Cl present in the factor profile by Zhang et al. (2013), Wang et al. (2008), Song et al. (2007) and Song
et al. (2006), in accordance with source profiles determined in the laboratory (Zheng et al., 2005).
High Cl associated with fine aerosols in winter is a distinctive feature in Beijing and even around
inland China, which is ascribed to coal combustion (Wang et al., 2008). Contrarily, Tian et al. (2016)
identified coal combustion based on a high contribution of OC and EC, while the high contribution
of Cl was attributed to a biomass burning source, similar to another study (Ma et al., 2017a). In other
studies (Li et al., 2019; Yu et al., 2013; Liu et al., 2019) coal combustion was resolved based on the
presence of metallic elements such as V, Se, Co, Cd, As and Ni, where V and Ni are widely used
markers for oil combustion (Mazzei et al., 2008). High loadings of As and Se have also been reported



as a typical source characteristic of coal combustion (Vejahati et al., 2010). Similar to coal
combustion, biomass burning was often characterised using the presence of K (Li et al., 2019; Tian
et al., 2016; Yu et al., 2013; Song et al., 2007; Song et al., 2006; Zhang et al., 2013; Liu et al., 2019;
Wang et al., 2008), a typical marker of biomass burning. Farming in Beijing's suburban districts has
been extensive in recent years. Burning of the crop remnants and fallen leaves by farmers in autumn
and winter results in the enhanced emissions of K. In addition, the contributions of Cl and Na were
also considered for the identification of these sources in some cases, depending on the species used
within the input (Song et al., 2007; Song et al., 2006; Tian et al., 2016). This highlights the fact that
none of the studies have used organic markers such as picene and levoglucosan which are very
specific to these combustion sources as discussed before, which may cause uncertainty in the resolved
sources. However, in the present study the use of organic markers played a key role in the
identification of these sources and their better apportionment. Despite this, some issues were observed
with these identified sources during winter due to extreme meteorological conditions as well as co-
emission of these aerosols at the same time, probably indicative of poor performance of the PMF
model under certain conditions.

Other important sources linked to traffic emissions, industrial activities and dust, are commonly
resolved among all the studies. The characterisation of these sources was predominantly based on the
metallic elements. For example, Zn, Cu, and Pb including sometimes EC were most often used to
characterise traffic emissions among all previous studies. Both Zn and Cu have been identified within
brake linings and tyre fragments (Thorpe and Harrison, 2008) and Pb has been used in the past within
gasoline as an anti-knock additive in China (Li et al., 2019). However, Cu and Zn can also serve as
indicators for industrial sources (Li et al., 2017; Yu et al., 2013). Other metallic elements (e.g., Sb,
Cr, Mn, K, Br and Ba) were also considered in certain cases to trace traffic emissions (Ma et al.,
2017a; Tian et al., 2016; Yu et al., 2013). However, a high contribution of Cr, Mn and sometimes Fe
to the given sources has also been attributed to industrial activities. Both Cr and Cr-containing


compounds are widely used in metallurgy, electroplating, pigment, leather and other industries
(Dall'Osto et al., 2013). A previous study found that ferrous metallurgy could emit Mn (Querol et al.,
2006). Furthermore, both Fe and Mn are characteristic components of iron and steel industry
emissions. In addition, Co, Mg, Al, Ca, Cd, Pb, Tl, Zn, V, Ni and Cu were also considered for the
apportionment of industrial sources (Tian et al., 2016; Yu et al., 2013). Zhang et al. (2013) identified
a mixed source of traffic and incineration emissions, based on high loading of Cu, Zn, Cd, Pb, Sb,
Sn, Mo, $NO_3$ and EC. In the present study, the assignment of road traffic emissions was based on
high loadings of Zn and Pb. It was also seen that the given source may contain some influence from
industrial activities, as the industrial contribution was not resolved like previous studies and probably
accounted in other factors. Thus, it is clear that these metals could belong to several sources and their
proper assignment to respective sources is difficult in the complex environment.

The same issue was observed with the assignments of dust type (dust/road dust/soil dust/mineral
dust/yellow dust/local dust) sources. Although the dust type sources were often found to be composed
of crustal elements (e.g., Ca, Mg, Si, Ti, Al, Fe), the attribution of crustal elements to a particular
source was not consistent from one study to another previously. The two dust sources (road dust and
soil dust) identified in the present study also indicated mixing with other factors.

The identification of the secondary inorganic aerosol factors was often based on the high contribution
of water soluble ions ($NO_3^-$, $SO_4^{2-}$, $NH_4^+$), consistent with other studies (Ma et al., 2017a; Song et al.,
2007; Song et al., 2006; Tian et al., 2016; Liu et al., 2019; Wang et al., 2008; Zhang et al., 2013).
These results highlight the role of chemical species used in characterising source profiles and their
influence on the variability noticed in the Beijing metropolitan area. This issue arises because many
of these species are not source specific, making it challenging to directly link PMF factors to sources.
Pant and Harrison (2012), reviewing receptor modelling studies from India, noted a tendency to



attribute metal-rich source profiles to "industry" in a rather casual manner without evidence of local
industrial sources.

The change in sources and emissions over the course of time due to stringent emissions regulations
could also be considered plausible for the observed variability in the chemical profile and contribution
of identified sources. Li et al. (2019) showed levels of trace metals (V, Cr, Mn, As, Cd and Pb)
decreased more than 40% due to the emission regulations, while crustal elements decreased
considerably (4–45%), suggesting emissions from anthropogenic activities were suppressed. A
reduction in the contribution of sources such as dust and industrial activity was observed in the present
study and another recent study performed by Liu et al. (2019) relative to the previous ones, indicating
the effect of regulatory measures on the contribution of identified sources to $PM_{2.5}$. However, the
concentration of the majority of metallic elements (K, Cr, Mn, Fe, Co, Cu, Zn, As, Ag, Cd and Pb)
increased when pollution levels changed from clean days to heavily polluted days. This highlights
that specific atmospheric conditions could also play a major role for the observed variability. Another
factor is the time of year when these studies were conducted as some of the identified sources (e.g.,
coal combustion and biomass burning) exhibit typical seasonal patterns. During a low concentration
period, PMF models may have difficulty in resolving sources, leading to mixing of factors.

Overall, the present study provides a view of existing $PM_{2.5}$ sources in the Beijing metropolitan area
by applying the PMF model to a filter-based dataset, which included water soluble ions, metals and
organic markers. Despite this, factors that were resolved based on metal signatures were not fully
resolved and indicate a mixing of different sources. As a part of same campaign, also discussed above,
Liu et al. (2019) used a similar approach by applying PMF on high resolution (1-hour) data, which
included OC, EC, ions, and metals, and did not encounter any issue. However, previous filter based
PMF studies conducted in the Beijing region that mostly included ions and metals in their input
dataset often showed difficulty in the proper assignment of metals to their respective sources. Even
the use of metal signatures from one to another study was not consistent. This highlights that the low





temporal resolution of filter data could not capture fast occurring atmospheric processes in Beijing,
and may lead to a "blurring" of sources by the long averaging period. Atmospheric circulation and
dynamic mechanisms play a key role in persistent haze events in Beijing during the cold period (Wu
et al., 2017; Feng et al., 2014). Such events are associated with the high pollution periods and will
offer opportunities for chemical and physical transformation within the aerosol that lead to
contravention of the requirement of receptor models for preservation of chemical profiles between
source and receptor.

**4.    CONCLUSION**
This study presents the outcomes of PMF performed on the combined dataset collected at two sites
(IAP and PG) in the Beijing metropolitan area, including their comparison with source apportionment
results from other approaches or based on different measurements. The PMF analysis resulted in the
identification of seven sources: coal combustion, biomass burning, oil combustion, secondary
inorganics, traffic emissions, road dust and soil dust. These results were in a good agreement with
previously published source apportionment results made using PMF.  However, factors that were
resolved based on metal signatures were not fully resolved and indicate an internal mixing of different
sources. In particular, soil dust, road dust and some industrial sources have many elements in common
and are very difficult to distinguish.
PMF results were compared with sources resolved from CMB and with PMF performed on other
measurements (online AMS, offline AMS). Results showed a broad agreement with some notable
exceptions. While this study provides some confirmatory evidence on $PM_{2.5}$ source apportionment in
Beijing, it highlights weaknesses of the PMF method when applied in this locality, and the results
should be viewed in the context of studies using other methods such as CMB which appear able to
give a more comprehensive view of the key sources affecting air quality. No industrial source profiles



were used as inputs to the CMB model reported here, so CMB offers no further insights into possible
contributions from industry.

**AUTHOR CONTRIBUTIONS**
This study was conceived by ZS and RMH. DS performed the PMF analysis and wrote the paper with
the help of Z.S. and R.M.H. T.V.V. and D.L. conducted the aerosol sampling and laboratory-based
chemical analyses. X.W. and J.X. conducted the CMB modelling at PG and IAP sites, respectively.
All authors discussed the paper and approved the final version for publication.

**COMPETING INTERESTS**
The authors declare that they have no conflict of interest.

**SPECIAL ISSUE STATEMENT**
This article is part of the special issue "In-depth study of air pollution sources and processes within
Beijing and its surrounding region (APHH-Beijing) (ACP/AMT inter-journal SI)". It is not associated
with a conference.

**FINANCIAL SUPPORT**
This research has been supported by the Natural Environment Research Council (APHH and SOA
grants): NE/N007190/1 (AIRPOLL-Beijing), NE/S006699/1 (SOA).





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





**TABLE LEGENDS**

**Table 1.**     List of the studies conducted in the Beijing metropolitan area to evaluate PM sources.

**FIGURE LEGENDS**

**Figure 1.**     Factor profiles for identified factors at IAP and PG. The bars show the composition profile (left axis) and the dots, the Explained Variation (right axis).

**Figure 2.**     Temporal variation of identified factors at IAP and PG. Solid and broken lines represent IAP and PG, respectively.

**Figure 3.**     Contribution of different sources to $PM_{2.5}$ mass at IAP and PG.

**Figure 4.**     Correlations observed between PMF and CMB results at IAP. *If two outlying points are removed from the coal combustion-PMF, correlations are markedly improved.

**Figure 5.**     Correlations observed between PMF and CMB results at PG.

**Figure 6.**     Correlations observed between PMF and online AMS PMF results at IAP (winter). *If two outlying points are removed from the coal combustion-PMF, correlations are markedly improved.

**Figure 7.**     Correlations observed between PMF and offline AMS PMF results at IAP (winter).





**Table 1.** List of the studies conducted in the Beijing metropolitan area to evaluate PM sources.

| Reported Studies | PM size fraction | Sampling site | Study period | Identified PMF factors (% contribution) | | | | | | Input species |
| --- | --- | --- | --- | --- | --- | --- | --- | --- | --- | --- |
| | | | | Dust/soil dust*/road dust$^S$/mineral dust$^\alpha$/local$^\beta$/non-local$^\infty$ | Traffic/fossil fuel$^\pi$ | Coal combustion | Biomass burning | Secondary inorganics | Industrial | |
| Li et al. (2019) | PM$_{2.5}$ | Urban- IAP | 15$^{th}$ Sep – 12$^{th}$ Nov 2014 | - | - | - | - | - | - | Mg, Al, K, Ca and Fe, V, Cr, Mn, Co, Cu, Zn, Ag, Cd, Pb and As |
| | | Suburban- UCAS | 15$^{th}$ Sep – 12$^{th}$ Nov 2014 | - | - | - | - | - | - | |
| Liu et al. (2019) | PM$_{2.5}$ | PKU | Nov-Dec 2016 | 5 | 18 | 16 | 9 | 44 | 8 | OC, EC, NO$_3^-$, SO$_4^{2-}$, NH$_4^+$, Cl$^-$, Na$^+$, Mg, Al, K, Ca, Ba, Cr, Mn, Fe, Ni, Cu, Zn, As, Se, and Pb |
| Ma et al. (2017a) | PM$_{2.5}$ | Urban- IAP | 24$^{th}$ Feb - 12$^{th}$ Mar 2014 | 10* | 6 | 18$^\epsilon$ | 18$^\epsilon$ | 46 | 20 | Na$^+$, K$^+$, Mg$^{2+}$, Ca$^{2+}$, NO$_3^-$, SO$_4^{2-}$, NH$_4^+$, Cl$^-$, Al, Fe, Ti, Mn, Cu, Zn, Sb, Pb, Cr, PM2.5, EC, OC |
| Tian el al. (2016) | PM$_{2.1}$ | Urban- IAP | Mar 2013 – 28$^{th}$ Feb 2014 | 8.4$^S$ | 19.6 | 17.7 | 11.1 | 25.1 | 12.1 | Na, Mg, Al, K, Ca, Mn, Fe, Co, Ni, Cu, Zn, Mo, Cd, Ba, Tl, Pb, Th, U, Na$^+$, NH$_4^+$, K$^+$, Mg$^{2+}$, Ca$^{2+}$, Cl$^-$, SO$_4^{2-}$, NO$_3^-$, OC and EC. |
| | PM$_{2.1-9}$ | | Mar 2013 – 28$^{th}$ Feb 2014 | 10.9$^S$, 22.6$^\alpha$ | - | 7.8 | 11.8 | 9.8 | 5.1 | |
| Yu et al. (2013) | PM$_{2.5}$ | Urban-BNU | 1$^{st}$ Jan – 31$^{st}$ Dec 2010 | 12.7$^S$, 10.4* | 17.1, 16$^\pi$ | | 11.2 | 26.5$^\epsilon$ | 6 | Mg, Al, Si, P, S, Cl, K, Ca, Ti, V, Cr, Mn, Fe, Ni, Cu, Zn, As, Se, Br, Ba and Pb |
| Zhang et al. (2013) | PM$_{2.5}$ | PKU | April, July, Oct 2009 and Jan 2010 | 16* | 3$^+$ | 14 | 13 | 26 | 28 | Na$^+$, K$^+$, Mg$^{2+}$, Ca$^{2+}$, NO$_3^-$, SO$_4^{2-}$, NH$_4^+$, Cl$^-$, Al, Fe, Na, Mg, K, Ca, Ba, Co, Mo, Cd, Sn, As, Se, Rb, Ti, Mn, Cu, Zn, Sb, Pb, Cr, PM$_{2.5}$, EC, OC |
| Wang et al. (2008) | PM$_{2.5}$ | Urban-BNU | Summer and winter | 8.8$^\beta$, 6.7$^\infty$ | 5.9 | 16.7 | 11.8 | 12.7$^\epsilon$, 14.7$^¥$ | 8.8 | Na, K$^+$, Mg$^{2+}$, NO$_3^-$, SO$_4^{2-}$, NH$_4^+$, Cl$^-$, |





| Reference | PM | Site | Period | | | | | | | Species |
|---|---|---|---|---|---|---|---|---|---|---|
| | | | 2001 to 2006 | | | | | | | Al, Fe, Na, Mg, K, Ca, Co, Cd, As, Ti, Mn, Cu, Zn, Sb, Pb, S, Cr, BC, OC, $C_2O_4^{2-}$ |
| | $PM_{10}$ | | Summer and winter 2001 to 2006 | $23^{\beta}$ | 8.4 | 13.3 | 10.2 | 18.9 | 14.9 | |
| | $PM_{2.5}$ | Duolun[©] | Summer and winter 2001 to 2006 | $36.2^{\beta}$, $23.1^{\infty}$ | - | - | 15.6 | $7.1^{¥}$ | - | |
| | $PM_{10}$ | | Summer and winter 2001 to 2006 | $61.7^{\beta}$, $11^{\infty}$ | - | - | 18.1 | 4.1 | - | |
| Song et al. (2007) | $PM_{2.5}$ | Urban-PKU, OLC, SJS, TZ, and LX; Rural-MT | Jan 2004 | $7.8^{S}$ | 8.5 | 40.6 | 16.4 | $9.2^{£}$, $10.5^{¥}$ | - | OC, EC, $NO_3^-$, $SO_4^{2-}$, $NH_4^+$, $Cl^-$, Na, Mg, Al, K, Ca, Ti, V, Cr, Mn, Fe, Ni, Cu, Zn, As, Se, and Pb |
| | | | Aug 2004 | $4.4^{S}$ | 7.8 | 5.9 | 6.7 | $12.6^{£}$, $4.2^{¥}$ | - | |
| Song et al. (2006) | $PM_{2.5}$ | OT, NB, BJ, XY, CH | 6-day intervals in Jan, Apr, Jul, and Oct 2000 | $9^{S}$ and yellow dust | 6 | 19 | 11 | $17^{£}$, $14^{¥}$ | 6 | OC, EC, $NO_3^-$, $SO_4^{2-}$, $NH_4^+$, Na, Al, Si, Cl, K, Ca, Ti, V, Cr, Mn, Fe, Ni, Cu, Zn, As, Se, Br, Pb, and Mg |

[+] Reported as traffic and waste incineration emissions; [£]Secondary sulphur/[¥]secondary nitrate; [π] Fossil
fuel; [€] Reported as combined coal and biomass burning contribution; [©] Background site; University
of Chinese Academy of Sciences-UCAS; Peking University-PKU; Beijing Normal University-BNU;
Ming Tombs -OT, the airport-NB, Beijing University-BJ, Dong Si EPB-XY, and Yong Le Dian-CH


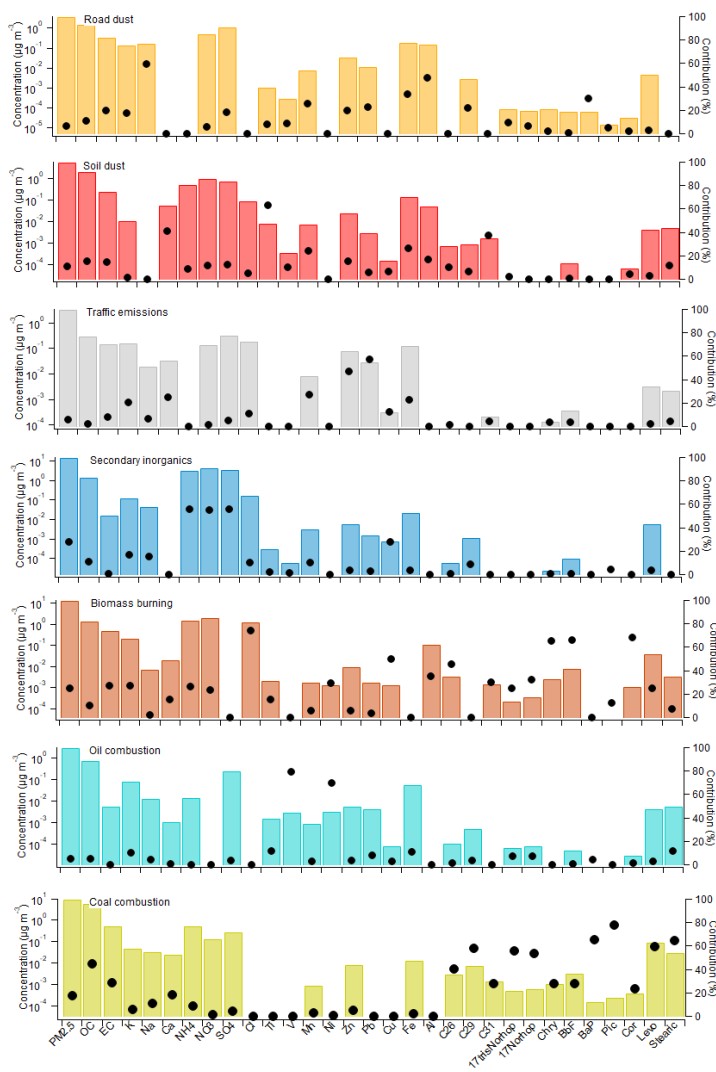


**Figure 1.** Factor profiles for identified factors at IAP and PG. The bars show the composition profile
(left axis) and the dots, the Explained Variation (right axis).






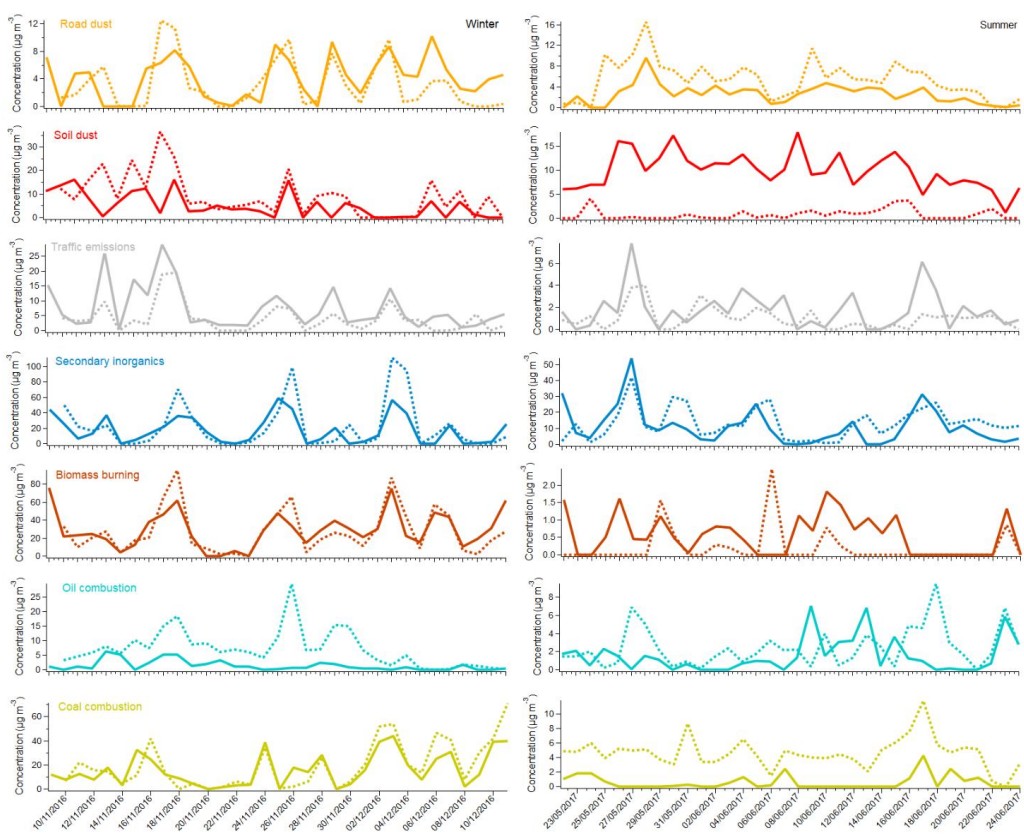

**Figure 2.** Temporal variation of identified factors at IAP and PG. Solid and broken lines represent IAP and PG, respectively.





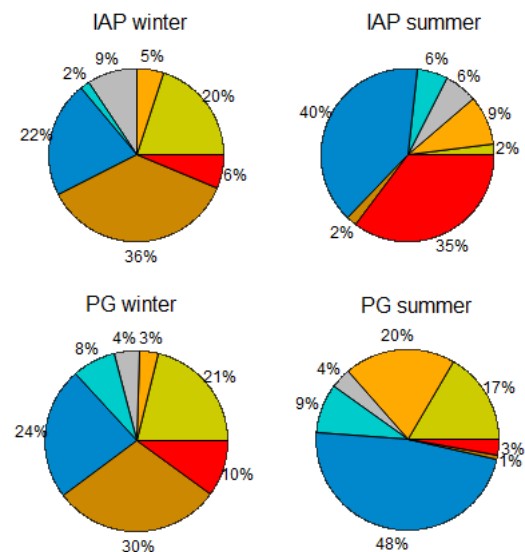

**Figure 3.** Contribution of different sources to PM$_{2.5}$ mass at IAP and PG.





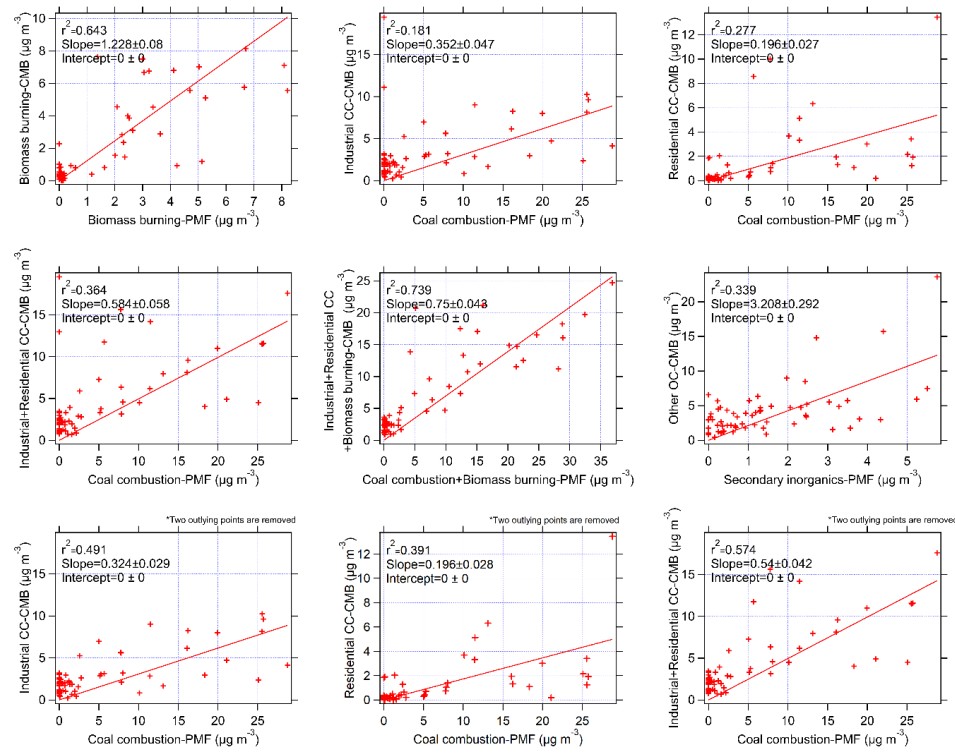


**Figure 4.** Correlations observed between PMF and CMB results at IAP. *If two outlying points are removed from the coal combustion-PMF, correlations are markedly improved.






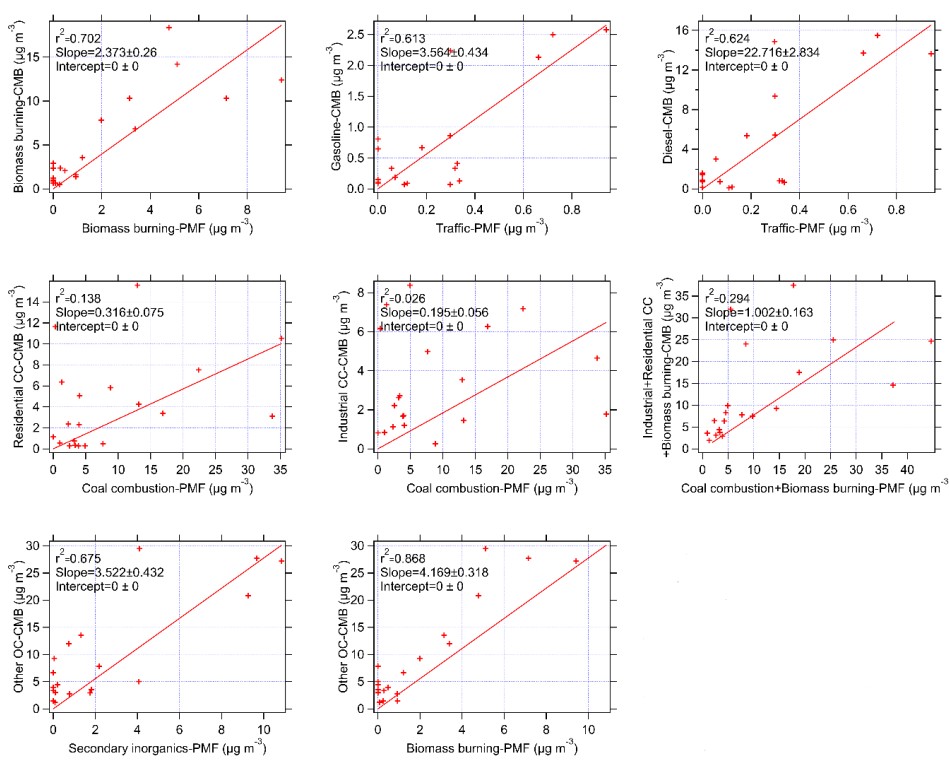


**Figure 5.** Correlations observed between PMF and CMB results at PG.



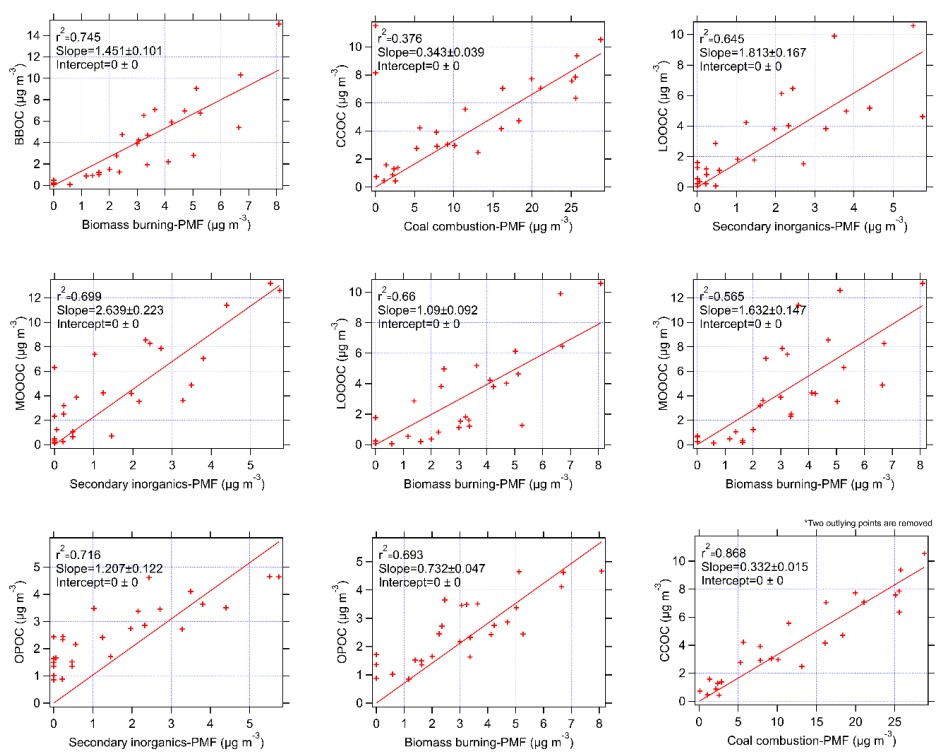


**Figure 6.** Correlations observed between PMF and online AMS PMF results at IAP (winter). *If two outlying points are removed from the coal combustion-PMF, correlations are markedly improved.







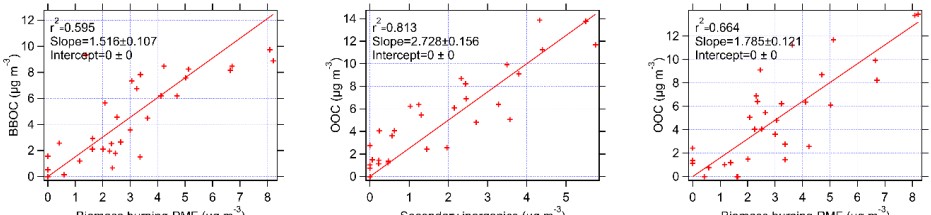


**Figure 7.** Correlations observed between PMF and offline AMS PMF results at IAP (winter).