# Peer review of "Insight into PM₂.₅ Sources by Applying Positive Matrix Factorization (PMF) at an Urban and Rural Site of Beijing"

_Atmospheric Chemistry and Physics, 2020_

## Author Response (AR1)

**Title: Insight into PM2.5 sources by applying Positive Matrix factorization (PMF) at an urban and rural site of Beijing**
**Author(s): Deepchandra Srivastava et al.**
**MS No.: acp-2020-1017**
**Special Issue: In-depth study of air pollution sources and processes within Beijing and its surrounding region (APHH-Beijing) (ACP/AMT inter-journal SI)**

**RESPONSE TO REVIEWERS**

**REVIEWER #1**

**Comment:** The discussion paper has been enhanced largely. Some comments are included below before publication.

**Response:** Firstly, we would like to thank the anonymous referee for their constructive comments, which we feel has added value to our manuscript. All comments have been considered and the manuscript has been amended accordingly. Amendments are in the track changes of the revised manuscript and supplementary information.

1. Lines 39-41. The contributions to PM2.5 in both IAP and PG are suggested to be given in specific values.

   **Response:** We have modified the text to include these data, as follows.

   Revised text (lines: 39-41): Major contributors to $PM_{2.5}$ mass were secondary inorganics (IAP: 22%; PG: 24%), biomass burning (IAP: 36%; PG: 30%), and coal combustion (IAP: 20%; PG: 21%) sources during the winter period at both sites.

2. Lines 465-466. I think soil dust has seasonal variation that high contribution in March due to the high wind speed and sandstorm.

   **Response:** We agree with this comment that there is a higher contribution of mineral dust in spring in Beijing due to higher wind speed and sandstorm. Our sampling period was May-June during the summertime and Nov-Dec during the wintertime. Unfortunately, we do not have spring data to show the dust effects. No clear seasonal variation was observed at PG, while a high contribution to $PM_{2.5}$ mass was noticed during the summertime at IAP.

   We have revised the text to make it clear.

   Revised text (489-491): No clear seasonal variation was observed at PG. However, this factor showed a high contribution (35%, 9.8 µg m$^{-3}$) to $PM_{2.5}$ mass during the summertime at IAP, while the contribution during other seasons at both sites was less than 10% (Figure 3).

3. The format of the paper still needs to be improved. Such as, lines 356-357, Cl$^-$; lines 703, NO$_3^-$

   **Response:** Has been corrected.

4. Line 380-382. The value in this work was much lower than the other published value (22-24% v.s. 44%). The authors should explain why so large difference were obtained, such as uncertainties of model, etc.

   **Response:** We agree with the concern raised here on the observed difference. Hence, we have revised the discussion.

Lines (380-390): Additionally, the factor showed a similar contribution (22-24%) to PM mass in winter at both sites. This value is lower than the value reported by Liu et al. (2019) at the other urban location (44%) in Beijing as a part of same APHH-Beijing campaign, although it should be noted that the sampling site and dates of sampling differed. We also noticed the source profile reported by Liu et al. (2019) contained a majority of all measured secondary inorganic species (>70%) as well as 20% of OC while the factor identified in the present study only accounted for ~55% of secondary inorganic species and 11% of OC with remaining fractions identified in other factors. Thus, although the identification of the factor was "secondary" in both studies, they do not represent exactly the same source. The modelled difference in the contribution of this factor to PM mass may also be related to the uncertainties of the input species: a filter-based dataset was used in the present study while Liu et al., (2019) used online measurements.

5. Line 427. "This factor makes a major contribution to crustal species, such as $Na^+$, Al and Fe". $Na^+$ is not the source marker of dust, why it got so high fraction in this factor? Maybe the uncertainty of the model?

    **Response:** We have added to the discussion about it.

    Added text (lines: 442-447): Such a high contribution of $Na^+$ in the identified factor was unexpected. Na is a major element of sea salt, sea-spray and marine aerosols (Viana et al., 2008), and has also been found to be enriched in fine particulates from coal combustion (Takuwa et al., 2006). It is notable that a high proportion of $Na^+$ was attributed to road dust in a previous study conducted at the same urban site (Tian et al., 2016), and a crustal source seems likely, but has not been confirmed.

6. Line 457-458. "urban, winter: r2 = 0.51" 0.51 is moderate high.

    **Response:** Yes, we agree. We have mentioned about the good correlation obtained between the estimated crustal dust and road dust during both seasons at PG and IAP. Please refer to the lines:

    Lines 474-478: A good correlation was observed between the estimated crustal dust and this factor during both seasons at PG (rural, winter: $r^2 = 0.78$, m (slope) =0.9; summer: $r^2 = 0.94$, m=0.5) and IAP (urban, winter: $r^2 = 0.51$, m=1.3; summer: $r^2 = 0.68$, m=1.2), highlighting that this may also contain a significant fraction of crustal dust (Figure S8). This suggests that the identified factor is not resolved cleanly and contains a mixed characteristic of road dust and crustal dust.

7. Line 490-491. "The use of Si in PMF could provide a better understanding on these dust related sources. However, it is not used in the present PMF input due to high number of missing data points" As discussion by the authors, Si is an important marker for dust. Missing Si might lead to underestimate the contribution of dust. It should be mentioned in the main text.

    **Response:** As we mentioned in the paper, Si was omitted due to a high number of missing points. However, we also noticed a good correlation between Si and Al (r2>0.9) and they both have been used extensively in source apportionment studies to trace dust related sources. Hence, it is unlikely that omission of Si from the PMF input will lead to underestimation of the contribution of dust related sources because the factor contribution to the mass of total PM is not dependent on a particular species, such as Si. In addition, it is also a common practice in the PMF model to use only one or two representative species for a source to limit the number of input species when they are well correlated

(Srivastava et al ., 2018, https://doi.org/10.1016/j.scitotenv.2017.12.135), and that should not affect source apportionment results.

8. 599-600 "This suggests the oxygenated fractions from AMS and secondary inorganics are subject to similar controls in the atmosphere". The authors should present more discussion for how to obtain this conclusion.

**Response:** We have added to the discussion to address this point.

Added lines (618-623): The formation of both secondary inorganic aerosol and oxygenated organic aerosol is dependent upon largely the same set of oxidant species, notably but not solely the hydroxyl and nitrate radicals. In both cases there are also both homogeneous and heterogeneous (aqueous phase) pathways, so conditions which promote the formation of oxidised organic aerosol will also favour formation of secondary inorganic aerosol, and hence a correlation is to be expected, and is often observed (Hu et al., 2016; Zhang et al., 2011).

706-707 "Thus, it is clear that these metals could belong to several sources and their proper assignment to respective sources is difficult in the complex environment". I agree with the authors that the metals maybe relate to other sources (factors), not the industrial activities.

**Response:** Thanks for this comment.

9. Figure 3. The figure should be revised. It is not clear to march the sources.

**Response:** The figure has been revised to provide greater clarity.

**REVIEWER #2**

**Comment:** The authors answered the questions well. And the paper has improved largely. The paper can be accepted after minor revision.

**Response:** Firstly, we would like to thank the anonymous referee for their constructive comments, which we feel has added value to our manuscript. All comments have been considered and the manuscript has been amended accordingly. Amendments are in the track changes of the revised manuscript and supplementary information.

1. Table 1 can be moved to SI.

**Response:** Table has been moved.

2. Fig. 3, the colours should be modified to better distinguish the sources.

**Response:** The figure has been revised to improve clarity.